# Low-Cost and Affordable Thermistor-Based Wideband Sub-THz Detector with Dielectric Waveguide Coupling

**DOI:** 10.3390/s24237533

**Published:** 2024-11-26

**Authors:** Przemysław Zagrajek, Marcin Wojciechowski, Paweł Komorowski, Kateryna Hovorova, Marcin Maciejewski

**Affiliations:** 1Institute of Optoelectronics, Military University of Technology, 2 Kaliski Street, 00-908 Warsaw, Poland; pawel.komorowski@wat.edu.pl (P.K.); marcin.maciejewski@wat.edu.pl (M.M.); 2Central Office of Measures, 2 Elektoralna Street, 00-139 Warsaw, Poland; marcin.wojciechowski@gum.gov.pl (M.W.); kateryna.hovorova@gum.gov.pl (K.H.)

**Keywords:** THz, sub-THz, detection, bolometer, quasi-optics

## Abstract

Bolometric detection of electromagnetic radiation is a well-established method in a wide frequency range, from millimeter waves through the terahertz region up to infrared. Fabrication of such a detector is often an expensive and demanding process. We propose a simple device based on a commercially available thermistor as a sensing element. To direct radiation to the sensor, we designed and fabricated a 3D-printed optical element integrated with the dielectric waveguide. An electronic setup was prepared to measure the sensor response. The described device is an affordable detector with acceptable detection parameters such as SNR or responsivity at a hundreds of volts per watt level.

## 1. Introduction

Terahertz radiation (THz) is still an important and interesting topic that pushes the boundaries of our understanding of nature [1]. Bolometric detectors of terahertz radiation play a key role in developing technologies for detecting electromagnetic radiation with frequencies ranging from 0.1 to 10 THz. The bolometer itself is not a new device. It was invented in 1878 by the astronomer Samuel Langley [2], but is still being developed and widely used in many applications. Detectors of this type, which operate by detecting temperature changes caused by radiation absorption, offer high sensitivity, making them ideal for applications in fundamental sciences such as spectroscopy, imaging, and astronomy [3,4,5].

The literature often refers to the wide use of bolometers, especially those based on superconducting materials [6] and silicon bolometers [7]. Studies on superconducting bolometers (Superconducting Transition Edge Sensors, TES) have shown that they are particularly effective in detecting very low-energy THz radiation due to their low operating temperature and high sensitivity. The use of TES has enabled significant progress in THz astronomy, especially in the observation of cosmic background radiation [8,9].

Thermistors, which are an important group of bolometric sensors, have also found their place in THz applications. The literature [10] describes their use for the precise measurement of small temperature changes, which allows the effective detection of low-power THz radiation. Thermistor-based bolometers detect resistance changes caused by an increase in temperature after radiation absorption. Popular materials used in such devices include semiconductor compounds and classical metals with high thermal conductivity, allowing for high-precision measurements. The THz range is not the only one in which thermistor-based detectors were applied. Such devices have been used successfully for many years to detect infrared radiation [11,12,13].

In addition to traditional bolometers, much modern research focuses on developing micro- and nano-sized bolometers with higher spatial resolution and faster response times [14,15,16,17,18]. Developing these technologies is an important contribution to a better understanding of the properties of THz radiation and its potential application. Research on various materials and detector designs is constantly evolving to meet the growing demands of science and technology [19].

However, it should be noted that creating one’s own superconducting or semiconductor-based sensor from scratch is not an easy task. Fabrication of micro-, nanostructures, or MEMS devices requires a specialized laboratory. In addition, integrating this structure with a metalized planar antenna, a dielectric resonator antenna, or any other antenna type used in this band with a single sensor is very demanding. We propose a solution based on commercially available, low-cost thermistors for users who cannot do it themselves and who do not need cutting-edge technology with record-breaking parameters. Because of the lack of an antenna, the device is wideband and does not depend on the polarization direction.

Similarly, our experience [20,21] of installing the sensor directly on a silicon lens is also challenging. Additionally, such a lens is relatively expensive and shows a higher Fresnel reflectance than the plastic lens because of the large refractive index of silicon. Moreover, forming the lens with 3D printing is much cheaper and more affordable nowadays. This method gives the designer considerable flexibility to integrate with subsequent components. The ordinariness of this technique does not mean a lack of precision in the fabrication of elements, which is also necessary.

Finally, it should be mentioned that all operational amplifiers (with minimized thermal drift and low noise) were also selected from a commercial offer to construct the device.

In the next section, the device will be described in the following order: the idea in general, the bolometric sensor, the optical part, and the readout. In the last section, the results obtained from this instrument are presented.

To the best of our knowledge, commercially available thermistors have not been reported in the literature as bolometric sensors of THz/sub-THz radiation. In this work, we introduce a novel, low-cost solution for terahertz detection. A significant advantage of our detector is its ability to operate at room temperature, making it highly practical for a wide range of applications. Additionally, the detector is characterized by its ease of fabrication and affordability, requiring no sophisticated measurement equipment for its operation.

The innovation presented in this paper also lies in the development of a fully 3D-printed radiation-coupling dielectric opto/waveguide solution. This coupling structure efficiently collects radiation from free space and directs it to the active region of the detector. Importantly, this coupling mechanism is not limited to our bolometric detector; it can also be employed with other types of THz sensors, greatly enhancing its versatility and potential for integration with a variety of detection technologies.

The proposed device is not only straightforward to construct but also, despite its relatively long response time, demonstrates performance parameters comparable to those of other room-temperature detector architectures.

## 2. Device

### 2.1. General Idea

The bolometer, in principle, consists of an absorber, a thermometer isolated from an external environment, and a readout system. According to the principles of bolometric detection described in [3], when the absorbent element and the thermometer are localized in one element, this type of sensor is said to be monolithic, and this is our case.

Our device, as is often practiced, has two thermally separated thermistors. The radiation is delivered to one of them, while the second works as a reference. It allows to eliminate the influence of environmental temperature changes. Common practice is to mount these two temperature transducers as two legs of the Wheatstone Bridge. Chronologically, the first piece of equipment used to measure small changes in resistance was a precise galvanometer. A newer method of such measurement is to use separated current sources for each branch of the bridge and a precise digital voltmeter. The idea of bolometric detection is presented in Figure 1, where t_0_ is the initial temperature of the thermistors, t_THz_ is the increase of temperature caused by terahertz radiation, I_z_ is the rated current of the current sources, and U_z_ is the rated bias voltage for both sources.

For the first measurements, we used the mentioned setup with diode current stabilizer 1N5288 with I_z_ = 0.390 mA and minimal dynamic impedance equal to 4.10 MΩ biased by U_z_ = 25 V.

### 2.2. Thermistors

We decided to prepare the bolometer on the basis of thermistor-type temperature conversion. Four Negative Temperature Coefficient Thermistors (NTC) with a resistance equal to 10 kΩ (at 25 °C) were considered active elements. The symbols, manufacturers, dissipation factors, and relative Temperature Coefficients of Resistance (TCR) of the NTCs are listed below.

The value of TCR shows how much resistance changes while the temperature of the thermistor changes. TCR is expressed as
(1)$$\alpha =\frac{1}{R}\cdot \frac{dR}{dT}\;,$$
where α is TCR, *R* is resistance, *dR* is resistance change, and *dT* is temperature change.

Figure 2 shows a photograph of thermistors with dissipation factors ranging from 1 to 5 mW/K, arranged in the order in Table 1. These are miniature electronic components with the largest dimension in the order of 3 mm. The first three have a cuboidal structure, and the last one is spherical. Their active area is of the order of a few square millimeters. The flat surface of the first three allows good adhesion to an absorber or waveguide made of an insulating material. The spherical one has a higher sensitivity but can only be used for a metal waveguide or can be accommodated in a circular hole made in the absorber.

Assuming an average value of 3 mW/K for the 0.3 mW power delivered to the sensor, its temperature should decrease by approximately 0.1 K. Such a temperature increase should be easily detected as the voltage change, *dV*.
(2)$$dV=I_{Z}\cdot dR=I_{Z}\cdot R\cdot \alpha \cdot dT,$$

The expected signal for the above values is equal to 7.8 mV. At the same time, the current *I_z_* flows through and increases the resistance by a few K. This is an additional reason to use the Wheatstone Bridge with a reference thermistor, which eliminates this effect.

To confirm the usefulness of the chosen thermistors and compare the influence of incident radiation on their resistance, a simple test with direct thermistors illumination was performed. The obvious unknown was the absorption process by thermistors. All of the thermistors were used without an additional absorption layer. Electromagnetic radiation was absorbed directly by the NTC. A simple experimental setup was prepared to verify how the resistance changes with THz radiation. Each thermistor was placed in front of the 100 GHz emitter while the resistance was measured. The power density in the measurement plane was about 2 mW/mm^2^. During the measurement, they were additionally isolated from the external influence of visible and infrared radiation and airflow. The results of the resistance change during the turning of the radiation on and off are presented in the Figure 3 below.

As is visible, each thermistor changes resistance as a result of radiation absorption and the resulting increase in temperature. For the amount of absorbed power mentioned above, the rise time is approximately 300–400 s depending on the thermistor. The response time in the final configuration is even longer, probably due to the poorer connection between the two heat sinks.

Although 192103LETA01 showed the highest response, its spherical shape made it unsuitable for connecting to the waveguide. The NHQ103B375T10 thermistor was used in the following bolometer construction.

### 2.3. Radiation Coupling

The thermistors were placed inside a screening aluminum box to avoid electromagnetic radiation and an isolating box to avoid heating by contact. The external housing – visible in the Figure 4a, was fabricated with specially designed air filling to improve isolating properties. Instead of full wall filling, it was printed with only 15% filling. The gyroid infill pattern was applied. Additionally, inside the box, the insulation (also between the thermistors) was applied.

A unique optical and guiding setup was designed to deliver THz radiation to the sensor. Due to the high 3D printing accuracy, we chose this method to fabricate the radiation coupling setup. A dielectric waveguide (DW) was applied to minimize thermal contact with the sensor. The absorbance and refractive index of the tested materials were measured [22], and the Cyclic Olefin Copolymer (COC) filament was chosen as DW. Polypropylene (PP) was used as a mechanical support and DW cladding due to the lower value of the refractive index compared to that of COC. The DW and its shell were integrated with a hyperhemispherical lens made of PP to couple THz radiation from open space into DW. Such a design not only allows for the coupling of radiation from the free space into the waveguide but also ensures confinement of the radiation within the DW and its vicinity based on the DW guiding principle. COC was chosen as the target material for the DW core due to the exceptionally low absorption coefficient within the spectral range (0.061 cm^−1^ at 140 GHz and 0.500 cm^−1^ at 1 THz). On the other hand, PP has been chosen for cladding, as it is one of the unique polymers with a refractive index lower than COC that is still reasonably transparent (below 1 cm^−1^ at 600 GHz). Moreover, both materials are commercially available as filaments for FDM 3D printing.

The FDM 3D printer used in this experiment is characterized with a vertical accuracy of 2 µm and a horizontal accuracy of 10 µm. The resolution of the 3D printed structure in the vertical direction results from the layer thickness, which was set to 100 µm. Horizontal resolution is limited by the nozzle of the printer; in this case, it was 0.25 mm. However, the actual resolution of the smooth print is smaller than that and lies somewhere between the movement accuracy of the nozzle and its size. It comes from the fact that the extruded material spills onto the sides, effectively smoothing the step-like cross section of the printed element, which is beneficial in the case of smooth designs (such as the proposed hemispherical lens and tube).

Comparing the resolution to the wavelength of the radiation (from 3 mm at 100 GHz to 0.46 mm at 650 GHz), one can see that in all cases the structures are manufactured with a subwavelength resolution. Naturally, it can be expected that the performance of the coupler will decay with frequency (both because of the absorption in the material and because of the accuracy of the printing with respect to the wavelength). The diffractive lens manufactured with the same FDM 3D printing method and COC filament has been demonstrated to operate at frequencies as high as 1 THz [23]. Keeping in mind that hemisphere is a much simpler design than diffractive lens, one can expect the FDM 3D printing to be applicable at least up to this frequency.

It can be shown that in the paraxial approximation, the spherical interface fulfills the following imaging equation:(3)$$\frac{n_{air}}{x}+\frac{n_{mat}}{y}=\frac{n_{mat}-n_{air}}{r},$$
where $n_{mat}$ and $n_{air}$ are refractive indices of the material of the lens and surrounding (in this case, air), respectively; x is the distance between the imaged object and the lens interface (in the air); y is the distance between the lens interface and the image (within the lens); and *r* is the radius of the curvature of the spherical interface. Assuming illumination with a plane wave (x=∞), one can obtain the following formula for the focal length (f) of the lens:(4)$$f=\frac{n_{mat}\cdot r}{n_{mat}-n_{air}},$$
which, assuming $n_{mat}=1.48$, $n_{air}=1.00,$ and r=5 mm, results in f=15.4 mm.

As visible in Figure 5 (DW—yellow; DW cladding and lens—violet), the diameter of the lens was 10 mm and the distance from lens front to DW was 15.4 mm.

The active area of the resistor is equal to 3.4 mm^2^ (1.7 × 2 mm). The area of the lens used as a coupler is equal to 78.5 mm^2^. Even by reducing the active area to the central half of the lens, one obtains a tenfold increase in the amount of gathered radiation. The coupling efficiency of the hemispherical waveguide lens can be estimated to be above 50% [24]. Next, the absorption losses in the 3 cm long filament will vary between 17% at 100 GHz and 78% at 1 THz. Therefore, it can be estimated that the power density delivered to the resistor is very similar to the ambient one at 1 THz and up to five times higher at 100 GHz. The absorptivity of the resistor is assumed to be relatively high, although from the perspective of coupling performance of the waveguide device it does not influence the results (it is the same with and without the device).

Moreover, it should be noted that the proposed dielectric waveguide system provides additional shielding from thermal or scattered radiation, which will illuminate the device at a higher angle (which will not be coupled). This effect provides an additional improvement in the signal-to-noise ratio of the whole system.

### 2.4. Measuring Electronics

The measuring device is engineered to achieve precise and reliable measurements of electromagnetic power within the terahertz frequency range. It is specifically designed to amplify weak signals from thermistors, ensuring high measurement accuracy while maintaining stable and noise-resistant performance. The device consists of several key components that work together to process and enhance the detected signals.

The operating principle of the described measuring device is based on generating a stable current to bias a differential pair of temperature transducers, converting the temperature difference into a voltage, and amplifying this voltage to a level that allows the measurement signal to be recorded by a computer-based system. The electrical measuring device is described in detail in [25]. Below is a brief description.

The device comprises six modules: a DC reference voltage source, two identical voltage/current converters, a differential amplifier, an output amplifier, and a power supply. The core of the device includes a stable reference voltage source, which provides a consistent baseline for all measurements. This reference is crucial for the proper functioning of the signal amplification process, which converts voltage to current to supply temperature sensors that detect changes in terahertz radiation. The voltage-to-current conversion is carefully managed to maintain stability and accuracy.

Compared to the previous design, current sources were upgraded to increase the measurement precision. The diode current stabilizers were replaced by new ones made of a reference voltage source and a measuring operational amplifier to which the bolometer feedback loop will be connected. The voltage source with thermal drift of 0.05 ppm/K, ultra-low noise 1.2 μV p-p, and the MAX4425 operational amplifier with 5.9 nV/√Hz were used. Precision resistors were essential components, with a resistance of 100 kΩ (0.01% tolerance) and a temperature coefficient equal to 2 ppm/K.

Signal amplification is achieved through a multi-stage process, primarily focusing on maximizing gain while minimizing noise and distortion. The device uses differential amplification techniques, which allow for the enhancement of very low-power signals detected from terahertz radiation. The so-called low noise zero-drift differential amplifier was used to read the signal from the bridge. To ensure the precision of the measurements, the device is designed with a controlled bandwidth, which limits the range of frequencies that can pass through to reduce unwanted noise. This selective filtering contributes to the overall stability of the signal output, ensuring that the amplified signals remain within a desirable range for further analysis and measurement.

## 3. Experiment

### 3.1. Setup

The detection parameters of the prepared bolometer were investigated in an experimental setup shown in the Figure 6. A frequency multiplier based on Schottky diodes, manufactured by Virginia Diodes, Inc., Charlottesville, VA, USA was used as a radiation source. The beam was focused with an elliptical mirror onto the detector, and the power was monitored using an Erickson PM4 power meter, from Virginia Diodes, Inc., Charlottesville, VA, USA. The detector voltage signal was acquired with the PXIe-6124 DAQ, National Instruments, Austin, TX, USA.

Due to the wideband range of detection, the low-pass filter was applied in front of the detector. The cut-off frequency (−3 dB) was set to 1.5 THz. The measurements were realized at three different frequencies: 100 GHz, 270 GHz, and 650 GHz. The power of the radiation directed to the detector was equal to 12 mW, 0.5 mW, and 0.7 mW, respectively.

### 3.2. Results

The first measurement performed was to check the time for photoresponse stabilization. The results shown in Figure 7 demonstrate that the detector detects radiation but also has a long response time. The bolometer needed more than 70 min to return to the initial state after turning the emitter on and off.

Due to the long response time, we decided to modulate the radiation. The full response was not achieved then, but the experiments were a bit faster. The modulation frequency of 10 mHz was a compromise between the experiment time and the strength of the response. Modulation of incident beam amplitude with 10 mHz decreased detector response about 3 times. The results obtained for each radiation frequency show that our detector can be used in a wide frequency range. This is due to the omission of frequency-dependent elements, such as antennas. Nevertheless, a decrease in the photoresponse is slightly visible. The difference is probably caused by the higher absorption coefficient of both materials (COC and PP) at higher frequencies.

Taking into account the power of the incident beam, the responsivity of the bolometric detector was calculated for various frequencies. In Table 2, the responsivity and signal-to-noise ratio (SNR) are presented. The SNR was calculated as the ratio of the mean value to the standard deviation of the measured signal. For our device, it was about 1 mV. The responsivity would be higher for a longer time constant, but the measurement would be prolonged.

As visible, the described detector has a responsivity of several hundred V/W in the measured frequency range. However, a decrease in this parameter is visible with increasing frequency. The reason for this is the increase in the absorption coefficient of the dielectric material. It means that the device is wideband, but its responsivity is expected to be at least 4 times worse at 1 THz (78 V/W) than at 100 GHz.

In Table 3, we compare the detection parameters of our detector (named in Table 3 as “Bolometer”) with commercially available detectors: THz 10 from SLT GmbH [26] and THZ5I-BL-BNC and THZ12D-3S-VP-D0 from Gentec-EO [27]. The SLT THz 10 detector works in combination with the current preamplifier. The responsivity of this detector is about 60 V/W for a 10^8^ V/A preamplifier setting but could be amplified an additional 100 times. The amplification was chosen to obtain a comparable SNR (about 150 V/V) for incident power equal to 0.5 mW (270 GHz). The maximum power that could be detected by the bolometric detector is calculated as the power necessary to heat the thermistor to 150 °C, which is the maximal operating value. The parameters for the bolometer were chosen for 270 GHz radiation frequency (0.5 mW). The value of responsivity is different from Table 2 due to the longer measurement time (815 s instead of 100 s like in Figure 8).

As is visible, the described detector presents detection parameters comparable to those of the other commercial devices. The quantity that is clearly different is the rise time. All detectors presented have advantages and disadvantages resulting from their properties. For a rapidly changing signal, the first device seems to be suitable; however, its responsivity is not the highest. For small signal detection, the second device seems to be favorable, but it could not work with high power, contrary to the third device, which is more durable but has a longer response time than both previous devices and cannot detect low-power signals. The last device in Table 3, which is the subject of this paper, has the longest response time but can also be used to detect low and high power signals.

## 4. Conclusions

There is still a lack of cheap instruments that allow to work with terahertz radiation. The various commercially available thermistors were tested as bolometric sensors. This kind of sensor is easy to use, and its integration does not require special laboratory or sophisticated equipment. The opto/waveguide system for coupling the radiation with the sensor was designed, 3D printed, and successfully used in the detector. Such a solution is a much simpler way of radiation delivery to the sensor than a combination with the silicon lens or within the metallic waveguide. The practical, effective, and high-stability readout system was fabricated on the basis of inexpensive electronic elements.

We proposed a cost-effective bolometer detector for sub-THz radiation. It is a wideband device with a slow response time, but its responsiveness allows for detecting sub-terahertz radiation in many typical applications. In future research, the authors plan to improve the response time of the device. Upgraded thermal connections are likely to result in an improvement in this matter. Another solution is to add an absorber to increase the heating of the thermistor.

The advantage of our detector is the measurement head, which contains only dielectric coupling elements and thermistors separated from the electronic elements. It means that it could be used in non-friendly environments where typical electronic devices could not work properly. Furthermore, it is much more durable and therefore resistant to electrical discharges than microelectronic devices such as Schottky diodes.

## Figures and Tables

**Figure 1 sensors-24-07533-f001:**
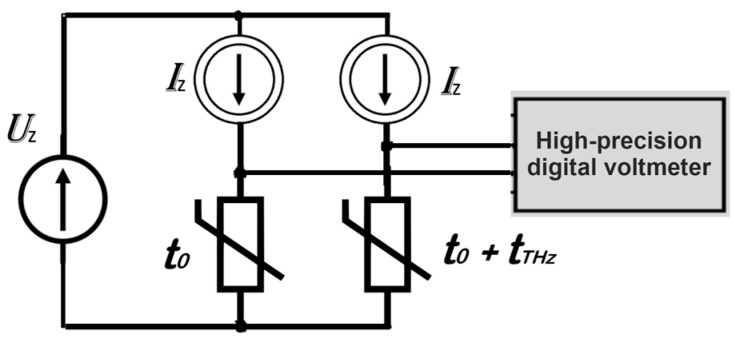
The idea of measuring the signal from a pair of thermistors. Two stabilized current sources, I_z_, are biased from one source, U_z_. Both thermistors are at initial temperature t_0_. The temperature of one thermospherical conductor increases by t_THz_ as a result of the absorption of electromagnetic radiation.

**Figure 2 sensors-24-07533-f002:**
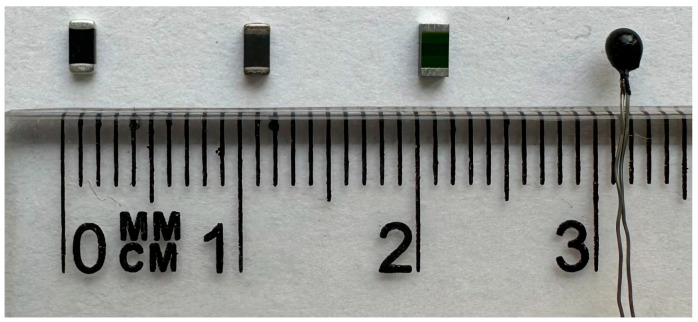
Photo of the thermistors shown with a millimeter scale to visualize the size of the thermistors. Consecutively, from the left: B57621C5103062, NC20K00103MBA, NHQ103B375T10, 192103LETA01.

**Figure 3 sensors-24-07533-f003:**
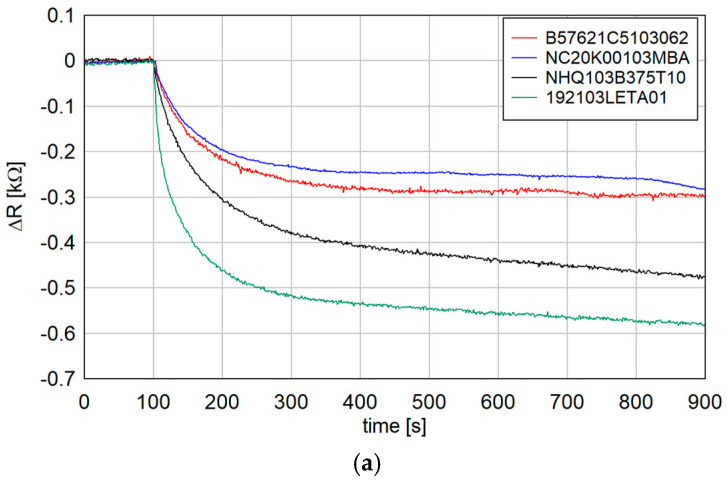
(**a**) Resistance change after turning on the illumination, and (**b**) resistance change after turning off the illumination.

**Figure 4 sensors-24-07533-f004:**
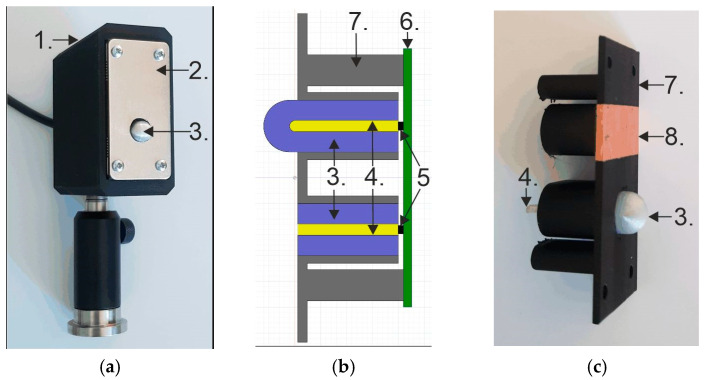
(**a**) General view of the detector: 1. External insulation housing; 2. Front plate of the screening shield; 3. Front of the collecting lens; Internal thermal isolation is not visible. (**b**) Scheme of the opto/waveguide part of the detector: 3. Lens and waveguide cladding; 4. Dielectric waveguide; 5. Thermistors; 6. PCB; 7. Supporting frame. (**c**) View of internal parts of the detector: 3. Lens and waveguide cladding; 4. Dielectric waveguide; 7. Supporting frame; 8. Internal metallic tape to screen reference thermistor.

**Figure 5 sensors-24-07533-f005:**
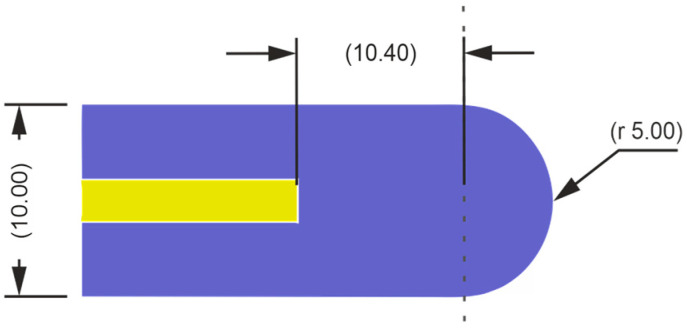
Details of the opto/waveguide part of the detector with dimensions in millimeters. The dielectric waveguide (1.8 mm in diameter) made of COC ($n_{COC}=1.51$) is marked in yellow, and the DW cladding integrated with the lens (10 mm in diameter) made of PP ($n_{PP}=1.48$) is marked in violet.

**Figure 6 sensors-24-07533-f006:**
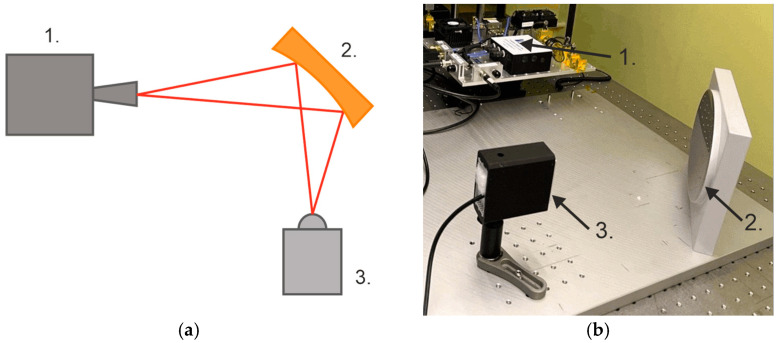
The experimental setup consists of a source of radiation (1), an elliptical mirror (2), and the detector under test (3). (**a**) Scheme, and (**b**) general view. The LP filter used to avoid heating the detector by the infrared is not shown.

**Figure 7 sensors-24-07533-f007:**
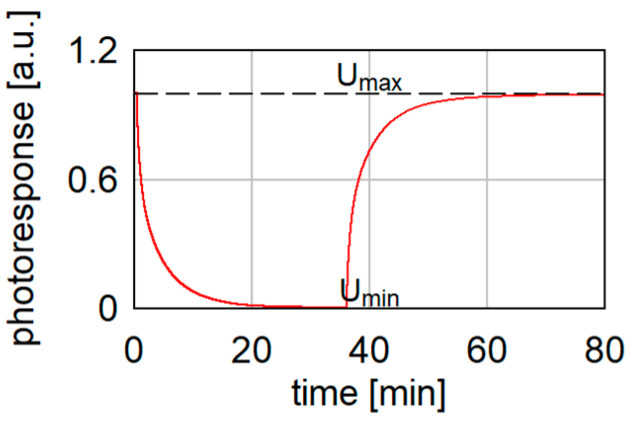
Normalized photoresponse of the detector during exposure to a beam of 100 GHz frequency and 12 mW power.

**Figure 8 sensors-24-07533-f008:**
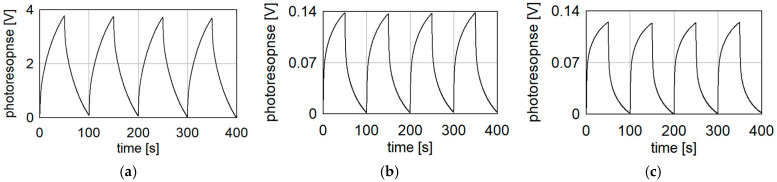
Results obtained with 10 mHz modulation for three frequencies of incident radiation: (**a**) 100 GHz; (**b**) 270 GHz; (**c**) 650 GHz.

**Table 1 sensors-24-07533-t001:** Thermistors parameters.

Symbol	Manufacturer	Dissipation Factor	TCR (at 25 °C)
B57621C5103062	TDK	5 mW/K	−3.8%/K
NC20K00103MBA	Kyocera	4 mW/K	−4.0%/K
NHQ103B375T10	Amphenol	3 mW/K	−3.8%/K
192103LETA01	Honeywell	1 mW/K	−4.4%/K

**Table 2 sensors-24-07533-t002:** Parameters of the detector.

Frequency [GHz]	Responsivity [V/W]	SNR [V/V]
100	312	3730
270	264	136
650	176	127

**Table 3 sensors-24-07533-t003:** Comparison with commercial detectors.

Device	Responsivity [V/W]	Maximum Power [mW]	Minimum Power [µW]	Rise Time [s]	Net Price [EUR]
THz 10	60	10	8	0.0001	3785
THZ5I-BL-BNC	70,000	0.625	0.1	0.2	3181
THZ12D-3S-VP-D0	0.2	3000	70–100	3	2462
Bolometer (this work)	530	600	4	815	<1000

## Data Availability

Data are contained within the article.

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
