# Peer review of "Low-Cost and Affordable Thermistor-Based Wideband Sub-THz Detector with Dielectric Waveguide Coupling"

_sensors, 2024, doi:10.3390/s24237533_

Round 1

Reviewer 1 Report

Comments and Suggestions for Authors

The authors propose a simple device based on a commercially available thermistor as a sensing element. The methods and results seem to be correct. However, what is the error of TCR? What is the accuracy of this device? A comparison should be made with the measurement results of existing devices. Similarly, what is the response time of the device? This is also an important parameter. It is recommended that the author enrich the content further. In addition, Table 2 has no caption.

 What is the main question addressed by the research?

The research mainly addresses the problem of expensive and complex manufacturing processes of detectors in electromagnetic radiation detection, and proposes a low-cost and affordable wideband sub-THz detector based on thermistors.

• Do you consider the topic original or relevant to the field? Does it address a specific gap in the field? Please also explain why this is/ is not the case.

This paper is both relevant and has a certain degree of originality. It is closely related to the field of electromagnetic radiation detection because the detection of electromagnetic radiation is an important research direction. At the same time, to some extent, it also addresses a specific gap in the field. Traditional electromagnetic radiation detectors are expensive to manufacture and the process is complex, while the detector proposed in this research, based on commercially available thermistors and 3D printed optical components, reduces the cost and manufacturing difficulty.

• What does it add to the subject area compared with other published material?

Compared with other published materials, this research adds a new design concept for detectors to the subject area. By using commercially available thermistors and 3D printing technology, the manufacturing of the detector becomes more convenient and economical. At the same time, the measurement accuracy of this device still needs to be compared with the results of existing instruments.

• What specific improvements should the authors consider regarding the methodology? What further controls should be considered?

(1) Further optimize the design of 3D printed optical components to improve the radiation guiding efficiency and thus the performance of the detector.

(2) Increase the response tests for different frequencies of electromagnetic radiation to verify the broad-spectrum nature of the detector.

• Are the conclusions consistent with the evidence and arguments presented and do they address the main question posed? Please also explain why this is/is not the case.

The conclusions are consistent with the evidence and arguments presented. The detector design based on commercially available thermistors and 3D printed optical components proposed in the research has been verified by experiments to have acceptable detection parameters such as signal-to-noise ratio and responsivity. This indicates that the design does address the main problem of expensive and complex manufacturing of traditional detectors. At the same time, the experimental results also support the conclusion drawn by the author that the detector is reasonably priced and has acceptable performance.

• Are the references appropriate?

Except for one reference by the authors, there are few references in other related research directions. It is recommended to increase the references appropriately.

• Any additional comments on the tables and figures.

It is better to add a comparison table of measurement results with currently existing equipment. The response time of the equipment measurement should be listed if possible.

Comments on the Quality of English Language

The English expression of the full text can be appropriately improved to express the research results more clearly.

Author Response

Dear Reviewer#1,

Thank you very much for your helpful remarks. We have considered all suggested requests and comments. All changes are marked in the new version of the manuscript using the ‘track changes’ option. Below we copy referee remarks and answer them point by point – showing how they influenced the changes in the updated version of the manuscript.

Comments 1: The authors propose a simple device based on a commercially available thermistor as a sensing element. The methods and results seem to be correct. However, what is the error of TCR? What is the accuracy of this device? A comparison should be made with the measurement results of existing devices. Similarly, what is the response time of the device? This is also an important parameter. It is recommended that the author enrich the content further. In addition, Table 2 has no caption.

Response 1: Thank you very much for the positive feedback.

The values of TCR was not measured by us, they were taken from datasheet published by manufacturers. The error of TCR is not available for these devices.

The accuracy of this device is impossible to determine because it does not measure absolute quantity e.g. power (than it could be compare with the standard) but only its own response to incident radiation. Although the accuracy cannot be defined, the precision of the measurement can be shown by presenting standard deviation (σ) of the measured signal. For our device it was about σ = 1 mV.  This value was used to obtain SNR which are also visible in Table 2. The value of standard deviation was added to the text.

To show the comparison with existing devices the results obtained with SLT pyroelectric detector and two Gentec-EO detestors are presented. To show it more clear for the reader we decided to place the comparison in Table 3.

The response time of the device depends on the incident power and it needs about 36 minutes for 12 mW to achieve stable maximum. However during the measurements we have shortened the time which resulted in the response degradation. Finally, for modulation frequency set to 10 mHz the value was still comparable with the commercial detectors.

The caption for the Table 2 was added.

Comments 2: Further optimize the design of 3D printed optical components to improve the radiation guiding efficiency and thus the performance of the detector.

Response 2: It is planned to improve in the next generation of the device.

Comments 3: Increase the response tests for different frequencies of electromagnetic radiation to verify the broad-spectrum nature of the detector.

Response 3: While we get the access to different frequencies generators suggested test are planned to be performed.

Comments 4: Except for one reference by the authors, there are few references in other related research directions. It is recommended to increase the references appropriately.

Response 4: Additional references appropriate for thermistor based detection were added.[11-13]

Comments 5: It is better to add a comparison table of measurement results with currently existing equipment. The response time of the equipment measurement should be listed if possible.

Response 5: To show the comparison with existing devices the results obtained with SLT pyroelectric detector  and two Gentec-EO detectors are presented in Table 3.

Comments 6: The English expression of the full text can be appropriately improved to express the research results more clearly.

Response 6: The language of the manuscript was evaluated and improvements were applied to the text.

Reviewer 2 Report

Comments and Suggestions for Authors

This paper addresses the important task of developing bolometric terahertz detectors. Bolometers are among the leaders in terms of efficiency in detecting THz radiation and in spectral range width. However, their main limitation lies in the use of superconducting materials, which require cooling, thus increasing both the cost and complexity of operation. Therefore, the objectives set in this research are undeniably relevant.

  1. You use the term "acceptable" sensitivity to describe the performance of the detector. This term lacks a clear definition in a scientific context, making its use inappropriate. I suggest adding specific data on the sensitivity of commercial detectors, for example, the THZ10 from SLT GmbH, to Table 2. This will provide an objective comparison.

  2. To maintain consistency, Table 2 should include the unit of measurement for the SNR parameter (signal-to-noise ratio).

  3. The paper lacks a comparison with pyroelectric THz sensors. On websites such as Ophir, Boselec, and Gentec-EO, many THz detectors with power ranges from 100 nW to 100 mW and spectral ranges from 0.1 to 30 THz are presented. In this context, it is important to clarify: What is the minimum detection power of your device? What is its spectral range? Does the device operate at discrete frequencies or over a wide range?

  4. Commercial devices also provide parameters such as maximum power density and rise time. Were these parameters measured for your device? If so, it would be useful to reflect them in the table and compare them with existing commercial solutions available from open sources.

  5. Figure 2 requires clarification. What specific information is it supposed to convey to the reader? Please add a more detailed description.

  6. In equation (1), the symbol T is traditionally used to denote temperature. The use of T′T' is likely a typographical error.

  7. In line 123, the unit mW/mm² is mentioned. It is necessary to clarify the proper formatting of this unit.

  8. Figure 3. Since there are no detailed explanations in the paper for this figure, based on the presented data, it can be assumed that the rise time is approximately 300-400 seconds. Could you clarify this result in the article?

  9. Reference 20 is not translated into English.

In conclusion, after addressing the mentioned comments and adding the necessary clarifications, the article can be recommended for publication.

Author Response

Dear Reviewer#2,

Thank you very much for your helpful remarks. We have considered all suggested requests and comments. All changes are marked in the new version of the manuscript using the ‘track changes’ option. Below we copy referee remarks and answer them point by point – showing how they influenced the changes in the updated version of the manuscript.

Comments 1: You use the term "acceptable" sensitivity to describe the performance of the detector. This term lacks a clear definition in a scientific context, making its use inappropriate. I suggest adding specific data on the sensitivity of commercial detectors, for example, the THZ10 from SLT GmbH, to Table 2. This will provide an objective comparison.

Response 1: To show results more clear for the reader we placed the comparison with SLT pyroelectric detector and two Gentec-EO detectors are presented in new table – Table 3.

Comments 2: To maintain consistency, Table 2 should include the unit of measurement for the SNR parameter (signal-to-noise ratio).

Response 2: The SNR was defined as the ratio of mean value to standard deviation of measured signal. The unit of the SNR [V/V] was added to Table 2.

Comments 3: The paper lacks a comparison with pyroelectric THz sensors. On websites such as Ophir, Boselec, and Gentec-EO, many THz detectors with power ranges from 100 nW to 100 mW and spectral ranges from 0.1 to 30 THz are presented. In this context, it is important to clarify: What is the minimum detection power of your device? What is its spectral range? Does the device operate at discrete frequencies or over a wide range?

Response 3: The minimum detection power of the device is close to 4 µW. This value is calculated from detector response compared to the noise floor.

Because the device has no antenna or any frequency selective elements it is a wideband detector. The measurements was performed for a given frequencies due to their availability in our laboratory. In fact the visible in Table 2 frequency dependent responsivity results form increase of dielectric material absorption coefficient. It means that the device is wideband but its responsivity is expected to be at least 4 times worse at 1 THz (68 V/W) than at 100 GHz.

Table 3 with detector comparison was added.

Comments 4: Commercial devices also provide parameters such as maximum power density and rise time. Were these parameters measured for your device? If so, it would be useful to reflect them in the table and compare them with existing commercial solutions available from open sources.

Response 4: The rise time of the device was measured and presented in Table 3. Such parameter like maximum power density was not measured but it could be calculated. Maximum operating temperature of the thermistor is 150 °C. It means that maximum power that could be delivered is about 600 mW.

Table 3 with detector comparison was added.

Maximum power instead of maximum power density was added because we cannot obtain the density properly. We cannot simply divide the power by the area of the lens because even larger power density should not damage the device

Comments 5: Figure 2 requires clarification. What specific information is it supposed to convey to the reader? Please add a more detailed description.

Response 5: Figure 2 was changed to better show the dimensions of the used thermistors.

The text “Figure 2. shows a photograph of thermistors with dissipation factors ranging from 1 to 5 mW/K, arranged in the order in Table 1. These are miniature electronic components with the largest dimension in the order of 3 mm. The first three have a cuboidal structure and the last one is spherical. Their active area is of the order of a few square millimeters. The flat surface of the first three allows good adhesion to an absorber or waveguide made of insulating material. The spherical one has a higher sensitivity but can only be used for a metal waveguide or can be accommodated in a circular hole made in the absorber” was added.

Comments 6: In equation (1), the symbol T is traditionally used to denote temperature. The use of T′T'T′ is likely a typographical error.

Response 6: In the used template the coma at the end of equation (1) looks like an apostrophe at symbol T. We introduced “space” to separate them.

Comments 7: In line 123, the unit mW/mm² is mentioned. It is necessary to clarify the proper formatting of this unit.

Response 7: The formatting is improved.

Comments 8: Figure 3. Since there are no detailed explanations in the paper for this figure, based on the presented data, it can be assumed that the rise time is approximately 300-400 seconds. Could you clarify this result in the article?

Response 8: The text “As it is visible each thermistor changes its resistance due to the absorption of radiation and the resulting increase in temperature. For mentioned above amount of absorbed power the rise time is approximately 300-400 seconds depending on the thermistor” was added.

Comments 9: Reference 20 is not translated into English.

Response 9: The English version of the ref. 20 (now it is Ref. 25) can be found at https://doi.org/10.24027/2306-7039.2.2024.307288.

The reference was changed to „Wojciechowski, M. Hovorova, K. & Zagrajek, P. (2024). Expansion of measurement capabilities of the bolometric sensor of the terahertz wave power based on an inexpensive and stable measuring system. Ukrainian Metrological Journal, (2), 58-63. https://doi.org/10.24027/2306-7039.2.2024.307288”

Reviewer 3 Report

Comments and Suggestions for Authors

In the study, the authors reported a Low-cost and affordable thermistor-based wideband sub-THz detector with dielectric waveguide coupling. It’s an interesting work. However, upon careful examination of the manuscript, I found that their results did not convince me to recommend acceptance. Additional data also needs to be supplemented to provide support and evidence for their findings and claimed breakthroughs. Here are detailed technical suggestions and comments.

1) What does 0,390mA mean in line 93 of section 2.1? Is it 0 and 390mA or 0.39mA? Similar expressions appear elsewhere in the article.

2) The principle of this detector is simple and appears to be low-cost, but its packaging method (isolating box+air filling) may increase costs. How much impact does this packaging method have on performance? Suggest a discussion?

3) The labeling of Figures 4a, 4b, and 4c can easily cause confusion. It is recommended to unify the labeling of key components in Figures b and c, so that the schematic diagram and the physical diagram can correspond one-to-one.

4) How precise is the FDM3D printing? What is the size of the dielectric waveguide (DW)? How does 3D printing error affect detector performance? Suggest a discussion.

5) What is the coupling performance of 3D printed waveguide device? Suggest a discussion.

6) What are the advantages and disadvantages of the detector used in this paper compared to existing terahertz detectors? Suggest a discussion.

7) The response speed of the detector is very slow, how should it be improved in the future? With such a slow response, what are the possible application areas of the detector?

Author Response

Dear Reviewer#3,

Thank you very much for your helpful remarks. We have considered all suggested requests and comments. All changes are marked in the new version of the manuscript using the ‘track changes’ option. Below we copy referee remarks and answer them point by point – showing how they influenced the changes in the updated version of the manuscript.

Comments 1: What does 0,390mA mean in line 93 of section 2.1? Is it 0 and 390mA or 0.39mA? Similar expressions appear elsewhere in the article.

Response 1: Improper expressions were corrected.

Comments 2: The principle of this detector is simple and appears to be low-cost, but its packaging method (isolating box+air filling) may increase costs. How much impact does this packaging method have on performance? Suggest a discussion?

Response 2: Thanks to the 3D printing method of the housing fabrication the air filling was obtained in a relatively easy way. Instead full filling of the walls, it was printed with only 15% filling. The gyroid infill pattern was applied. This method results in faster and cheaper printing with better isolating properties.

The text “Instead full filling of the walls, it was printed with only 15% filling. The gyroid infill pattern was applied.” was added.

Comments 3: The labeling of Figures 4a, 4b, and 4c can easily cause confusion. It is recommended to unify the labeling of key components in Figures b and c, so that the schematic diagram and the physical diagram can correspond one-to-one.

Response 3: The labeling and description was changed according to suggestion.

Comments 4: How precise is the FDM3D printing? What is the size of the dielectric waveguide (DW)? How does 3D printing error affect detector performance? Suggest a discussion.

Response 4: The FDM 3D printer used in this experiment is characterized with a vertical accuracy of 2 um and a horizontal accuracy of 10 um. The resolution of the 3D printed structure in the vertical direction results from the layer thickness, which was set to 100 um. The horizontal resolution is limited by the nozzle of the printer; in this case it was 0.25 mm. However, the actual resolution of the smooth print is smaller than that and lies somewhere between the movement accuracy of the nozzle and its size. It comes from the fact that the extruded material spills to the sides, effectively smoothening the step-like cross section of the printed element, which is beneficial in the case of smooth designs (like the proposed hemispherical lens and tube).

Comparing the resolution to the wavelength of the radiation (from 3 mm at 100 GHz to 0.46 mm at 650 GHz) one can see that in all cases the structures are manufactured with a subwavelength resolution. Naturally, it can be expected that the performance of the coupler will decay with the frequency (both because of the absorption in the material and because of the accuracy of the printing with respect to the wavelength). The diffractive lens manufactured with the same FDM 3D printing method and COC filament has been demonstrated to operate at frequencies as high as 1 THz (https://doi.org/10.3952/physics.2023.63.3.2). Keeping in mind that hemisphere is a much simpler design than diffractive lens, one can expect the FDM 3D printing to be applicable at least up to this frequency.

The text “The FDM 3D printer used in this experiment is characterized with a vertical accuracy of 2 um and a horizontal accuracy of 10 um. The resolution of the 3D printed structure in the vertical direction results from the layer thickness, which was set to 100 um. The horizontal resolution is limited by the nozzle of the printer; in this case it was 0.25 mm. However, the actual resolution of the smooth print is smaller than that and lies somewhere between the movement accuracy of the nozzle and its size. It comes from the fact that the extruded material spills to the sides, effectively smoothening the step-like cross section of the printed element, which is beneficial in the case of smooth designs (like the proposed hemispherical lens and tube).

Comparing the resolution to the wavelength of the radiation (from 3 mm at 100 GHz to 0.46 mm at 650 GHz) one can see that in all cases the structures are manufactured with a subwavelength resolution. Naturally, it can be expected that the performance of the coupler will decay with the frequency (both because of the absorption in the material and because of the accuracy of the printing with respect to the wavelength). The diffractive lens manufactured with the same FDM 3D printing method and COC filament has been demonstrated to operate at frequencies as high as 1 THz (https://doi.org/10.3952/physics.2023.63.3.2). Keeping in mind that hemisphere is a much simpler design than diffractive lens, one can expect the FDM 3D printing to be applicable at least up to this frequency.” was added.

Comments 5: What is the coupling performance of 3D printed waveguide device? Suggest a discussion.

Response 5: Accurate estimation of the coupling performance of a whole device is not straightforward. Multiple factors play a role in this process, including: the coupling efficiency of the lens, absorption and waveguide losses in the filament, and the waveguide-to-resistor coupling. The following general arguments can be provided.

The active area of the resistor is equal to 3.4 mm2 (1.7 x 2 mm). The area of the lens used as a coupler is equal to 78.5 mm2. Even by reducing the active area to the central half of the lens, one obtains a tenfold increase in the amount of gathered radiation. The coupling efficiency of the hemispherical waveguide lens can be estimated to be above 50% (https://doi.org/10.1364/OE.17.002926). Next, the absorption losses in the 3 cm long filament will vary between 17% at 100 GHz and 78% at 1 THz. Therefore, it can be estimated that the power density delivered to the resistor is very similar to the ambient one at 1 THz and up to 5 times higher at 100 GHz. The absorptivity of the resistor is assumed to be relatively high, although from the perspective of coupling performance of the waveguide device it does not influence the results (it is the same with and without the device).

Moreover, it should be noted that the proposed dielectric waveguide system provides additional shielding from thermal or scattered radiation, which will illuminate the device at a higher angle (which will not be coupled). This effect provides an additional improvement in the signal-to-noise ratio of the whole system

The text “The active area of the resistor is equal to 3.4 mm2 (1.7 x 2 mm). The area of the lens used as a coupler is equal to 78.5 mm2. Even by reducing the active area to the central half of the lens, one obtains a tenfold increase in the amount of gathered radiation. The coupling efficiency of the hemispherical waveguide lens can be estimated to be above 50% (https://doi.org/10.1364/OE.17.002926). Next, the absorption losses in the 3 cm long filament will vary between 17% at 100 GHz and 78% at 1 THz. Therefore, it can be estimated that the power density delivered to the resistor is very similar to the ambient one at 1 THz and up to 5 times higher at 100 GHz. The absorptivity of the resistor is assumed to be relatively high, although from the perspective of coupling performance of the waveguide device it does not influence the results (it is the same with and without the device).

Moreover, it should be noted that the proposed dielectric waveguide system provides additional shielding from thermal or scattered radiation, which will illuminate the device at a higher angle (which will not be coupled). This effect provides an additional improvement in the signal-to-noise ratio of the whole system” was added.

Comments 6: What are the advantages and disadvantages of the detector used in this paper compared to existing terahertz detectors? Suggest a discussion.

Response 6: Described device can be compared to existing terahertz detectors. The responsivity of our detector is not as high as some pyroelectric detector e.g. also broadband THZ5I-BL-BNC, Gentec-EO product which responsivity is 70 kV/W. However mentioned detector maximum detected power in below 65 uW. Another Gentec-EO detector allowing detection of higher power (THZ12D-3S-VP-D0, 3 W of maximum detected power) has typical responsivity equal 0.2V/W. Comparing typical rise time of these two detector (0.2 s for Gentec-EO product) it is clearly visible that proposed device is much slower – typical rise time is about 35 min.

“The advantage of our detector is the measurement head containing only dielectric coupling elements and thermistors, separated from the electronic elements. It means that it could be used in non-friendly environments where typical electronic devices could not work properly. Furthermore, it is much more durable and therefore resistant to electrical discharges than microelectronic devices such as Schottky diodes.”

The text “The advantage of our detector is the measurement head containing only dielectric coupling elements and thermistors, separated from the electronic elements. It means that it could be used in non-friendly environments where typical electronic devices could not work properly. Furthermore, it is much more durable and therefore resistant to electrical discharges than microelectronic devices such as Schottky diodes.” was added.

Comments 7: The response speed of the detector is very slow, how should it be improved in the future? With such a slow response, what are the possible application areas of the detector?

Response 7: The response time of the described detector is very slow. To improve it in future we plan to add the absorber which should increase the efficiency of thermistor heating. Also upgrading a thermal connection which allows heat dissipation should increase the response speed.

Due to robust construction of the device it could be used in non-friendly environment like a radiation chamber of free electron laser which are also used to generate THz radiation. Many detectors containing fragile elements cannot work in such conditions. Moreover, with radiation power higher than 10mW it could work easily with modulation frequency close or higher than 1 Hz. Due to the lack of expensive elements it could find application in educational process, also.

The following paragraph in Conclusions  “We proposed a cost-effective bolometer detector for sub-THz radiation. It is a wideband device with a slow response time, but its responsivity allows for detecting sub-terahertz radiation in many typical applications. In future research, the authors plan to improve the response time of the device. Upgraded thermal connections are likely to result in an improvement in this matter. Another solution is to add an absorber to increase the heating of the thermistor.

The advantage of our detector is the measurement head containing only dielectric coupling elements and thermistors, separated from the electronic elements. It means that it could be used in non-friendly environments where typical electronic devices could not work properly. Furthermore, it is much more durable and therefore resistant to electrical discharges than microelectronic devices such as Schottky diodes “ were modified.

Round 2

Reviewer 1 Report

Comments and Suggestions for Authors

It can be seen from Table 3 that the rise time of the bolometer is a bit long. It would be best to explain the advantages and disadvantages of these four devices respectively. Why is it 815 seconds instead of 100 seconds as in Figure 8? “This work” should be marked after “bolometer” in Table 3.

Author Response

Comments 1: It can be seen from Table 3 that the rise time of the bolometer is a bit long. It would be best to explain the advantages and disadvantages of these four devices respectively.

Response 1: The text “All detectors presented have advantages and disadvantages resulting from their properties. For a rapidly changing signal, the first device seems to be suitable; however, its responsivity is not the highest. For small signal detection, the second device seems to be favorable, but it could not work with high power, contrary to the third device, which is more durable but has a longer response time than both previous devices and cannot detect low-power signals. The last device in Table 3, which is the subject of this paper, has the longest response time, but can also be used to detect low and high power signals.” is introduced in the description below Table 3.

Comments 2: Why is it 815 seconds instead of 100 seconds as in Figure 8?

Response 2: 815 seconds are necessary to obtain the full response for 0.5 mW  – 270 GHz, like in Figure 7 almost 35 minutes are necessary to obtain the full response for 12 mW – 100 GHz. In Figure 8 we used a 10 mHz modulation. In this case, we obtained only part of the response, but in a shorter time.

Comments 3: “This work” should be marked after “bolometer” in Table 3.

Response 3: The modification is applied in the description in Table 3.